# DISE: Dynamic Integrator Selection to Minimize Forward-pass Time in Neural ODEs

## Abstract

Neural ordinary differential equations (Neural ODEs) are appreciated for their ability to significantly reduce the number of parameters when constructing a neural network. On the other hand, they are sometimes blamed for their long forward-pass inference time, which is incurred by solving integral problems. To improve the model accuracy, they rely on advanced solvers, such as the Dormand–Prince (DOPRI) method. To solve an integral problem, however, it requires at least tens (or sometimes thousands) of steps in many Neural ODE experiments. In this work, we propose to i) directly regularize the step size of DOPRI to make the forward-pass faster and ii) dynamically choose a simpler integrator than DOPRI for a carefully selected subset of input. Because it is not the case that every input requires the advanced integrator, we design an auxiliary neural network to choose an appropriate integrator given input to decrease the overall inference time without significantly sacrificing accuracy. We consider the Euler method, the fourth-order Runge–Kutta (RK4) method, and DOPRI as selection candidates. We found that 10-30% of cases can be solved with simple integrators in our experiments. Therefore, the overall number of functional evaluations (NFE) decreases up to 78% with improved accuracy.

## 1 Introduction

Neural ordinary differential equations (Neural ODEs) are to learn time-dependent physical dynamics describing continuous residual networks (Chen et al., 2018). It is well known that residual connections are numerically similar to the explicit Euler method, the simplest integrator to solve ODEs. In this regard, Neural ODEs are considered as a generalization of residual networks. In general, it is agreed by many researchers that Neural ODEs have two advantages and one disadvantage: i) Neural ODEs can sometimes reduce the required number of neural network parameters, e.g., (Pinckaers & Litjens, 2019), ii) Neural ODEs can interpret the neural network layer (or time) as a continuous variable and a hidden vector at an arbitrary layer can be calculated, iii) however, Neural ODEs's forward-pass inference can sometimes be numerically unstable (i.e., the underflow error of DOPRI's adaptive step size) and/or slow to solve an integral problem (i.e., too many steps in DOPRI) (Zhuang et al., 2020b; Finlay et al., 2020; Daulbaev et al., 2020; Quaglino et al., 2020).

Much work has been actively devoted to address the numerically unstable nature of solving integral problems. In this work, however, we are interested in addressing the problem of long forward-pass inference time. To overcome the challenge, we i) directly regularize the numerical errors of the Dormand–Prince (DOPRI) method (Dormand & Prince, 1980), which means we try to learn an ODE that can be *quickly* solved by DOPRI, and ii) dynamically select an appropriate integrator for each sample rather than relying on only one integrator. In many cases, Neural ODEs use DOPRI, one of the most advanced adaptive step integrator, for its best accuracy. However, our method allows that we rely on simpler integrators, such as the Euler method or the

Table 1: MNIST classification results (1 NFE ≈ 0.0007 seconds for a test batch of 100 images)

| Model | Accuracy | NFE |
|---|---|---|
| ResNet | 0.9959 | N/A |
| RKNet | 0.9953 | N/A |
| No reg. | 0.9960 | 26 |
| Kinetic energy reg. | **0.9965** | 20 |
| $L^1$ reg. | 0.9956 | 20 |
| $L^2$ reg. | 0.9961 | 20 |
| Our reg. | 0.9964 | 14 |
| No reg. & DISE | 0.9958 | 11.36 |
| Our reg. & DISE | 0.9963 | **5.79** |

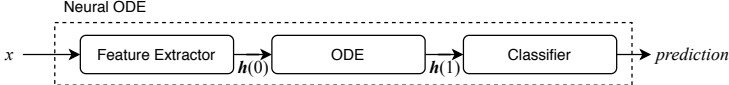

Figure 1: The general architecture of Neural ODEs. We assume a classification task in this figure.

fourth-order Runge–Kutta (RK4) method (Ixaru & Vanden Berghe, 2004), for carefully selected inputs.

Table 1 shows an experimental result that our proposed regularization not only reduces the number of function evaluations (NFE) — the inference time is linearly proportional to the number of function evaluations in Neural ODEs — but also increases the inference accuracy in the MNIST classification task. We can reduce the inference time by reducing the average number of steps (and thus, the average NFE) of DOPRI, which can be obtained when the learned ODE is trained to be in a suitable form to solve with DOPRI with a proper regularization.

However, the NFE of DOPRI in a step is 6 whereas RK4 has 4 and the Euler method has 1. So, the Euler method is six times faster than DOPRI even when their step sizes are identical. Therefore, the automatic step size adjustment of DOPRI is not enough to minimize the NFE of forward-pass inference (see Section B in Appendix for more detailed descriptions with a concrete example). To this end, we design an auxiliary network that chooses an appropriate integrator for each sample. The combination of our regularization and the proposed **D**ynamic **I**ntegrator **SE**lection (DISE) shows the best performance in the table.

We conduct experiments for three different tasks and datasets: MNIST image classification, PhysioNet mortality prediction, and continuous normalizing flows. Our method shows the best (or close to the best) accuracy with a much smaller NFE than state-of-the-art methods. Our contributions can be summarized as follows:

1. We design an effective regularization to reduce the number of function evaluations (NFE) of Neural ODEs.
2. We design a sample-wise dynamic integrator selection (DISE) method to further accelerate Neural ODEs without significantly sacrificing model accuracy.
3. We conduct in-depth analyses with three popular tasks of Neural ODEs.

## 2 RELATED WORK

In this section, we review the literature on Neural ODEs. In particular, we review recent regularization designs for Neuarl ODEs and numerical methods to solve ODEs.

### 2.1 NEURAL ODES

It had been attempted by several researchers to model neural networks as differential equations (Weinan, 2017; Ruthotto & Haber, 2019; Lu et al., 2018; Ciccone et al., 2018; Chen et al., 2018; Gholami et al., 2019). Among them, the seminal neural ordinary differential equations (Neural ODEs), as shown in Fig. 1, consist of three parts in general: a feature extractor, an ODE, and a classifier (Chen et al., 2018; Zhuang et al., 2020a). Given an input $x$, the feature extractor produces an input to the ODE, denoted $\boldsymbol{h}(0)$.

Let $\boldsymbol{h}(t)$ be a hidden vector at layer (or time) $t$ in the ODE part. In Neural ODEs, a neural network $f$ with a set of parameters, denoted $\boldsymbol{\theta}$, approximates $\frac{\partial \boldsymbol{h}(t)}{\partial t}$ and $\boldsymbol{h}(t_1)$ becomes $\boldsymbol{h}(0) + \int_{t_0}^{t_1} f(\boldsymbol{h}(t), t; \boldsymbol{\theta}) \, dt$, where $f(\boldsymbol{h}(t), t; \boldsymbol{\theta}) = \frac{\partial \boldsymbol{h}(t)}{\partial t}$. In other words, the internal dynamics of the hidden vector evolution is described by an ODE. One key advantage of Neural ODEs is that we can reduce the number of parameters without sacrificing model accuracy. For instance, one recent work based on a Neural ODE marked the best accuracy for medical image segmentation with an order of magnitude smaller parameter numbers (Pinckaers & Litjens, 2019). In general, we calculate

$\boldsymbol{h}(1)$[1] and feed it into the next classifier and its final prediction is made. One can accordingly modify the architecture in Fig. 1 for other types of tasks. For simplicity but without loss of generality, we assume the architecture in our discussion.

Neural ODEs have been used in many tasks, ranging from classification and regression to time series forecasting and generative models (Yildiz et al., 2019; Grathwohl et al., 2019; Rubanova et al., 2019).

## 2.2 ODE SOLVERS

DOPRI is one of the most powerful integrators (Hairer et al., 1993) and widely used in Neural ODEs. It is a member of the Runge–Kutta family of ODE solvers. DOPRI dynamically controls the step size while solving an integral problem. It is now the default method for MATLAB, GNU Octave, and Simulink. It internally estimates an error by using a heuristic method and the step size is determined by a function inversely proportional to the estimated error — the larger the error, the shorter the step size. The error at $i$-th step of DOPRI for an integral problem $x$, denoted $err_{x,i}$, is estimated by the difference between the fourth-order and the fifth-order Runge–Kutta methods at the moment. The intuition behind the heuristic error estimation is simple yet effective. Among simpler methods, we consider the Euler method, and the fourth-order Runge–Kutta (RK4) method. The Euler method is the simplest method to solve ODEs and both the Euler method and RK4 use a fixed step size. Therefore, their solving time is deterministic.

One step of DOPRI involves six function evaluations, i.e., six function calls of $f$. The Euler method calls the network $f$ only once in a step and RK4 calls four times. Therefore, the Euler method is six times faster than DOPRI for a step. The term 'NFE' refers to the number of function evaluations to solve an integral problem. For the Euler method and RK4, NFE is deterministic and does not vary. In DOPRI, however, NFE varies from one sample to another, depending on the estimated error and the number of steps. We refer readers to Section B in Appendix for more detailed descriptions with a concrete example.

## 2.3 REGULARIZATIONS IN NEURAL ODES

To make Neural ODEs faster, one possible way is regularizing the ODE function $f$. Two naïve methods are regularizing $\boldsymbol{\theta}$ with the $L^1$ or $L^2$ regularizers (Ng, 2004). Strictly speaking, these two regularizers are to prevent overfitting. Therefore, preventing overfitting does not necessarily mean quick forward-pass inference.

To this end, Dupont et al. showed that by augmenting $\boldsymbol{h}(t)$ with additional zeros, i.e., augmenting the dimensionality of $\boldsymbol{h}(t)$, one can achieve similar effects (Dupont et al., 2019). However, this method is meaningful when we cannot freely control the dimensionality of $\boldsymbol{h}(t)$, which is not our setting. Recently, a kinetic regularization concept has been proposed by Finlay et al. (Finlay et al., 2020), which is written as follows:

$$R_k \stackrel{\text{def}}{=} \int_{t_0}^{t_1} \| f(\boldsymbol{h}(t), t; \boldsymbol{\theta}) \|_2^2 \, dt. \tag{1}$$

Among all regularization terms designed so far, this kinetic regularization's goal is the closest to ours. It can encourage Neural ODEs to learn straight-line paths from $\boldsymbol{h}(t_0)$ to $\boldsymbol{h}(t_1)$.

## 3 PROPOSED METHOD

While enabling the design of compact models, Neural ODEs have one critical drawback that they require solving integral problems, for which many approximation methods have been proposed: the Euler method, RK4, and DOPRI, to name a few. Almost all of them are based on discretizing $t$ and converting an integral into a series of additions. In many cases, therefore, it requires a dense discretization, resulting in a long forward-pass inference time.

---

[1]For simplicity but without loss of generality, the time duration can be fixed into $t \in [0, 1]$. Any arbitrary length ODEs can be compressed into a unit time interval ODE. In some time series datasets, however, the final integral time $t_1$ is given in a sample. In such a case, $t_1$ is set to the sample time.

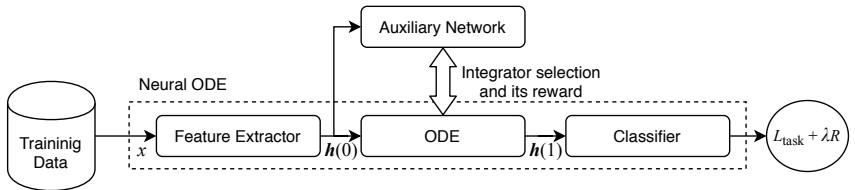

Figure 2: The overall workflow of our proposed method. We note that the Neural ODE and the auxiliary integrator selection network cooperate with each other.

In this paper, we tackle the problem of minimizing the number of function evaluations (and thereby, the forward-pass inference time) of Neural ODEs without significantly sacrificing model accuracy. Our proposed method consists of two parts: i) using the DOPRI's error estimation as a regularizer and ii) using an auxiliary network to select an appropriate integrator for each input sample.

### 3.1 DOPRI'S ERROR ESTIMATION AS A REGULARIZER

We re-implement the DOPRI method in PyTorch and make it return the estimated error terms. Let $\{err_{x,1}, err_{x,2}, \cdots, err_{x,N}\}$, where N is the number of steps of DOPRI, be an array of errors estimated by DOPRI while solving an integral problem for an input $x$. Note that the adaptive step size is an inverse function of the error at each step. We use the following regularizer while training Neural ODEs:

$$R_{err} \stackrel{\text{def}}{=} \sum_{x \in T} \sum_{i=1}^{N} err_{x,i}, \tag{2}$$

where $x$ is an input sample for which we have to solve an integral problem, and $T$ is a training set.

For instance, $x$ can be an image sample to classify. If we train a Neural ODE to classify images with the cross-entropy loss in conjunction with the regularizer, the trained Neural ODE will learn an ODE that can correctly classify images while reducing the forward-pass time of DOPRI.

The backward-pass calculation of our proposed regularizer can be done in $\mathcal{O}(\frac{1}{s_{avg}})$ by maintaining the forward-pass computation graph, where $s_{avg}$ is the average step size of DOPRI. However, this complexity will decrease as training goes on with our regularizer because the average step size will increase.

### 3.2 AUXILIARY NETWORK TO SELECT INTEGRATOR

We introduce our auxiliary network $v$ to dynamically select an appropriate integrator given input. This network $v$ cooperates with a Neural ODE as shown in Fig. 2. Before training the auxiliary network, we first train a target Neural ODE. We use the following loss function to train the target Neural ODE:

$$L_{task} + \lambda R, \tag{3}$$

where $L_{task}$ is a task-specific training loss, which can be optimized through the adjoint sensitivity method, and $R$ is an appropriate regularization term. $\lambda \geq 0$ is a coefficient to emphasize the regularization term.

We then train the auxiliary network $v(\boldsymbol{h}(0); \boldsymbol{\theta}_v)$ to predict the costs of the Euler method, RK4, and DOPRI. Given a fixed Neural ODE $f$ and $\boldsymbol{\theta}$, we use the same training data to collect the following data for each integrator:

$$c(\boldsymbol{h}(0); f, \boldsymbol{\theta}) \stackrel{\text{def}}{=} \begin{cases} \Delta^{\alpha}, & \text{if prediction is the same as ground-truth in the training data,} \\ M^{\alpha}, & \text{if otherwise,} \end{cases} \tag{4}$$

where $\Delta > 0$ is the number of function evaluations (NFE) to solve the integral problem of $\boldsymbol{h}(0) + \int_{t_0}^{t_1} f(\boldsymbol{h}(t), t; \boldsymbol{\theta}) \, dt$, $M$ is a large enough penalty, and $\alpha$ is an exponent. We evaluate this cost value

$c$ for each integrator and train the network $v$. The auxiliary network predicts the costs of the three integrators simultaneously, i.e., its output dimensionality is 3.

If the target task is not a classification problem, we use the following cost:

$$c(\boldsymbol{h}(0); f, \boldsymbol{\theta}) \stackrel{\text{def}}{=} \begin{cases} (\Delta\Gamma)^\alpha, \text{ if } \Gamma \leq \beta, \\ M^\alpha, \text{ if otherwise,} \end{cases} \tag{5}$$

where $\Gamma$ is an error estimation for a certain integrator, such as mean absolute/squared error, KL divergence, and so forth, for which we prefer small values. If larger values are preferred, the above definition should be accordingly modified. $\beta$ is a hyperparameter to decide a threshold.

Note that training $v$ becomes a regression problem with supervision. After many epochs, the auxiliary integrator selection network $v$ is stabilized and we can deploy it. The best integrator for $\boldsymbol{h}(0)$ is selected by finding the smallest predicted cost. We also note that we need to run the network $v$ only once to select an appropriate integrator for a test case, which incurs a little overhead.

The neural network architecture for $v$ should be carefully designed for each application. We introduce our general method in this section. In the experimental evaluation section, we will introduce our design choice for $v$ in each application.

## 4 EXPERIMENTAL EVALUATIONS

We describe our detailed experimental environments and results with the following three different tasks: i) MNIST image classification, ii) PhysioNet mortality prediction, and iii) continuous normalizing flows.

We conduct our experiments in the following sequence. We first compare the following regularization methods: i) Neural ODEs without any regularizer, ii) with the $L^1$ regularizer, iii) with the $L^2$ regularizer, iv) with the kinetic energy regularizer, and v) with our proposed regularizer. At this stage, we do not include the auxiliary network yet. This stage is to study which regularizer works better than others in each task. After selecting the best regularization method for each task, we train all networks including the auxiliary network.

Throughout these experiments, we show that i) the forward-pass of learned ODEs can be faster by using an appropriate regularizer, and ii) the auxiliary network strategically selects simpler integrators than DOPRI for carefully chosen input samples. For each experiment, we show NFE values and unit NFE time in seconds and its multiplication will be wall-clock time. We repeat training and testing with 10 different seeds and report the mean value in the main paper and the standard deviation value in Appendix. Experiments with various integrators are also in Appendix.

All experiments were conducted in the following software and hardware environments: UBUNTU 18.04 LTS, PYTHON 3.6.6, NUMPY 1.18.5, SCIPY 1.5, MATPLOTLIB 3.3.1, PYTORCH 1.2.0, CUDA 10.0, and NVIDIA Driver 417.22, i9 CPU, and NVIDIA RTX TITAN.

### 4.1 MNIST IMAGE CLASSIFICATION

We omit the description about MNIST. We use ODE-Net used in (Chen et al., 2018) to classify MNIST images. Refer to Appendix for detailed descriptions on ODE-Net with its hyperparameter configurations.

**Auxiliary Network Architecture.** In addition to ODE-Net, we have one more auxiliary network $v$ whose architecture is summarized in Table 2. We use the standard residual block (He et al., 2016) for this network and the proposed network consists of four layers. In comparison with the network $f$ of ODE-Net, its architecture is relatively simpler and it takes 0.8 NFEs (i.e., 0.0006 seconds) to run once. Recall in Table 1 that

Table 2: The auxiliary network for MNIST where $\sigma$ is ReLU, $\pi$ is Group Normalization, and $\xi$ is Adaptive Average Pooling.

| Layer | Design | In Size | Out Size |
|---|---|---|---|
| 1 | $\sigma$(Residual Block) | $64 \times 6 \times 6$ | $64 \times 6 \times 6$ |
| 2 | $\xi(\sigma(\pi(\text{Residual Block})))$ | $64 \times 6 \times 6$ | $64 \times 1 \times 1$ |
| 3 | FC after flattening | $64 \times 1 \times 1$ | $32 \times 1$ |
| 4 | FC | $32 \times 1$ | $3 \times 1$ |

the network (function) $f$ is evaluated 5-20 times on average to solve integral problems. Therefore, running the auxiliary network $v$ once to decide the best integrator can be a fruitful investment.

**Hyperparameters.** The regularization coefficient $\lambda$ is set to $\{0.01, 0.0001, 0.0001, 0.005\}$ and the starting learning rate is 0.1 with a decay factor $\{0.1, 0.01, 0.001\}$ at $\{60, 100, 140\}$ epochs. The exponent $\alpha$ is 0.3 and $M$ is 1,000. We use DOPRI as our default solver unless DISE is adopted. We use the recommended hyperparameters of ODE-Net in the paper or in the respective github repository, as noted in Appendix.

**Experimental Results.** The results are summarized in Table 1. It needs 26 NFEs when we do not use any regularizer to train ODE-Net. The kinetic energy-based regularizer improves both the accuracy and the NFE value. However, its NFE is comparable to other standard regularizations. Our regularizer shows an accuracy close to that of the kinetic regularization in the table with an NFE of 14, which is much faster than other cases.

We also applied the dynamic integrator selection (DISE) for both of the no-regularizer and our regularizer configurations. It is worth noting that the combination of our regularizer and the dynamic selection shows an extraordinary outcome in the table. Even after considering the overhead incurred by the auxiliary network, an NFE of 5.79 is very fast in comparison with other cases. Its accuracy is also larger than many other baselines.

Table 3: The distribution of the integrator selection by our auxiliary network for MNIST

| Integrator | Percentage |
|------------|------------|
| DOPRI | 37% |
| RK4 | 34% |
| Euler | 29% |

Table 3 shows a percentage of test cases where each integrator is selected by the auxiliary network with our regularizer. DOPRI occupies the biggest portion for its higher accuracy than others, i.e., 37%. RK4 and the Euler method have more numerical errors than DOPRI and as a result, their estimated costs are larger than that of DOPRI in general (due to the large penalty $M$). We note that their rankings in terms of the selection percentage are the same as those in terms of the numerical error. One interesting point is that the Euler method also occupies a non-trivial portion in the table.

## 4.2 PHYSIONET MORTALITY CLASSIFICATION

**Dataset.** We use the PhysioNet computing in cardiology challenge dataset released at 2012 (Silva et al., 2010). It is to forecast mortality rates in intensive care unit (ICU) populations. The dataset had been collected from 12,000 ICU stays. They remove short stays less than 48 hours and recorded up to 42 variables. Each records has a time stamp that indicates an elapsed time after admission to the ICU. Given a record, we predict whether the patient will die or not. Therefore, the task-specific loss $L_{task}$ in Eq. 3 is a cross-entropy loss. They well separated the dataset for training, validating, and testing.

Table 4: The auxiliary network for PhysioNet where $\psi$ is Drop Out.

| Layer | Design | In Size | Out Size |
|-------|--------|---------|----------|
| 1 | $\sigma$(FC) | $20 \times 1$ | $10 \times 1$ |
| 2 | $\sigma$(FC) | $10 \times 1$ | $10 \times 1$ |
| 3 | $\sigma$(FC) | $10 \times 1$ | $5 \times 1$ |
| 4 | $\psi(\sigma$(FC)) | $5 \times 1$ | $5 \times 1$ |
| 5 | FC | $5 \times 1$ | $3 \times 1$ |

We use Latent-ODE (Rubanova et al., 2019) for this task. The network (function) architecture of $f$ in Latent-ODE is described in Appendix in conjunction with its hyperparameter configurations.

**Auxiliary Network Architecture.** The auxiliary network for this dataset is shown in Table 4, which consists of four fully connected layers and an output layer. There is a dropout at the fourth layer and we found that using a dropout at this position improves the selection quality in some cases. The time to run the auxiliary network once is comparable to 0.023 NFE (i.e., 0.0003 seconds) in our testing, which is negligible.

Table 5: PhysioNet prediction results (1 NFE $\approx$ 0.013 seconds for a test batch of 60 records)

| Model | AUC | NFE |
|-------|-----|-----|
| No reg. | 0.7190 | 74 |
| Kinetic energy reg. | 0.7581 | 63.5 |
| $L^1$ reg. | **0.7630** | 68 |
| $L^2$ reg. | 0.7450 | 59 |
| Our reg. | 0.7509 | 39.71 |
| No reg. & DISE | 0.7513 | 57.57 |
| Our reg. & DISE | 0.7604 | **34.1** |

**Hyperparameters.** The coefficient $\lambda$ is set to $\{0.045, 0.0002, 0.0025, 0.015\}$ respectively, and the starting learning rate is 0.01 with a decay factor of

0.999. The exponent $\alpha$ is 0.05. The large penalty $M$ is 1,000. The dropout ratio is $\{0, 0.25\}$. We use DOPRI as our default solver unless DISE is adopted. We use the recommended hyperparameters for Latent-ODE in their paper or in their github repository.

**Experimental Results.** Table 5 summarizes all the results in terms of the AUC score and NFE. Adding an regularizer improves the AUC score and NFE at the same time in all cases. It is noted that the best AUC score is achieved when we use the $L^1$ regularizer for this task among all regularizers. Our regularizer shows the smallest NFE among all tested regularizers, i.e., an NFE of 74 with no regularizers vs. an NFE of 39.71 with our regularizer. When being used with DISE, our regularizer marks an NFE of 34.1 and an AUC score of 0.7604, which is the best AUC to NFE ratio in the table.

We summarized the distribution of the integrator selection by DISE with our regularizer in Table 6. For a majority of cases, it chooses DOPRI due to the difficult nature of the task. One interesting point is that RK4 had not been used at all because it does not show any notable difference from the Euler method in this task. In that regard, choosing the Euler method is a sensible decision.

Table 6: The distribution of the integrator selection by our auxiliary network for PhysioNet.

| Integrator | Percentage |
|---|---|
| DOPRI | 87.5% |
| RK4 | 0% |
| Euler | 12.5% |

### 4.3 Continuous Normalizing Flows

**Dataset.** Normalizing flows transform a simple distribution into a richer complex distribution, which can be used for many deep learning applications, such as generative models, reinforcement learning, and variational inference (Rezende & Mohamed, 2015). It is also known that Neural ODEs can generalize the concept of normalizing flows, which is called continuous normalizing flows. We use the experimental codes and data released in (Chen et al., 2018).

In this task, we perform a maximum likelihood training between $q(x)$ and $p(x)$, where $q$ is a distribution created by a flow model and $p$ is a target distribution. Therefore, the task-specific loss is to maximize $\mathbb{E}_{p(x)}[\log q(x)]$, i.e., minimize $L_{task} = -\mathbb{E}_{p(x)}[\log q(x)]$. The change of variable theorem is used to measure the probability $q(x)$. We set the initial distribution to a Gaussian distribution of

Table 7: The auxiliary network for Continuous Normalizing Flows. $\Phi$ means Tanh.

| Layer | Design | In Size | Out Size |
|---|---|---|---|
| 1 | (FC) | $32 \times 3$ | $32 \times 1$ |
| 2 | $\Phi$(FC) | $32 \times 1$ | $64 \times 1$ |
| 3 | $\Phi$(FC) | $64 \times 1$ | $32 \times 1$ |
| 4 | $\Phi$(FC) | $32 \times 1$ | $3 \times 1$ |

$\mathcal{N}(0, 1)$ and a flow model tries to transform it to the target distribution $p$. One good characteristic of continuous normalizing flows is that we can reverse the model and extract the reverse-mapping procedure from $q$ to the Gaussian as well.

**Auxiliary Network Architecture.** The auxiliary network for this dataset is shown in Table 7, which consists of three fully connected layers and an output layer. The time to run the auxiliary network once is comparable to 0.018 NFE (i.e., 0.000144 seconds) in our testing, which is negligible.

**Hyperparameters.** The coefficient $\lambda$ is set to $\{0.0005, 0.00001, 0.0003, 0.004\}$ and the learning rate is 0.001. The exponent $\alpha$ is 0.03. The large penalty $M$ is 1,000. We use DOPRI as our default solver unless DISE is adopted. We use the recom-

Table 8: Continuous Normalizing Flow results (1 NFE $\approx$ 0.008 seconds for a test batch of 32 samples)

| Model | NLP | NFE |
|---|---|---|
| No reg. | 0.8911 | 2297 |
| Kinetic energy reg. | 0.8914 | 2286 |
| $L^1$ reg. | 0.8901 | 2259 |
| $L^2$ reg. | 0.8760 | 2904 |
| Our reg. | 0.8841 | 2166 |
| No reg. & DISE | 0.8883 | 2104 |
| Our reg. & DISE | **0.8742** | **1984** |

mended hyperparameters for the base continuous normalizing flow model in the original paper or in their github repository. The threshold $\beta$ is set to the average negative log-probability loss of the case where we use our regularizer. The rationale behind this threshold is that we select the Euler method only when its loss value is good enough.

**Experimental Results.** Table 8 shows the performance of various methods. With no regularizers, the base Neural ODE model's NFE is very large, i.e., an NFE of 2,297. Our regularizer decreases it to 2,166 and the combination of our regularizer and DISE further decreases to 1,984. However, the $L^2$ regularizer rather increases the NFE and has a negative influence on it. In almost all cases, adding

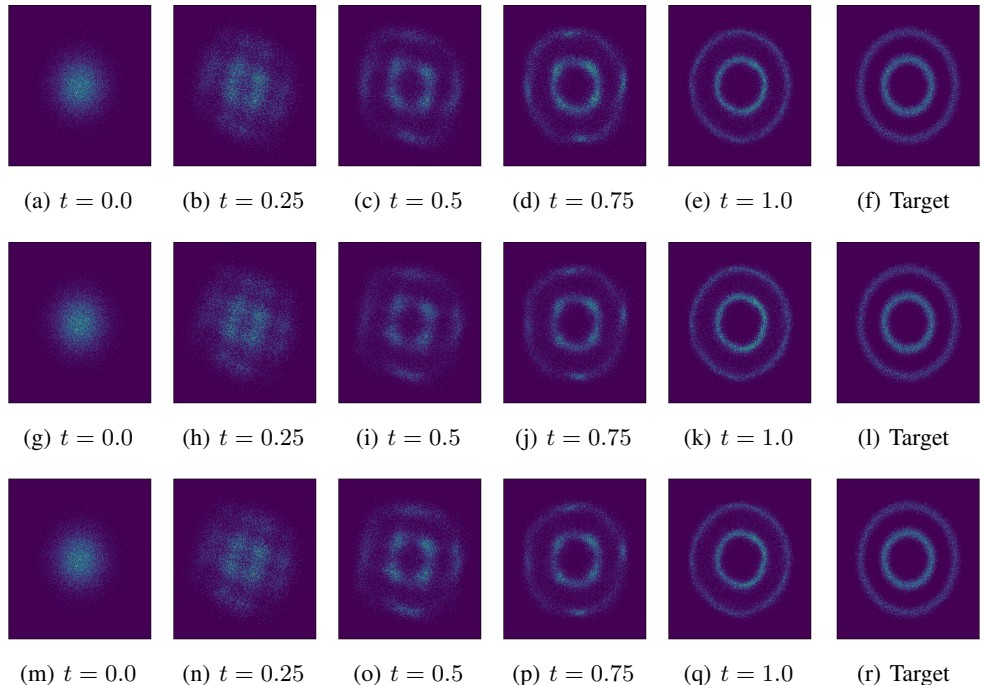

(a) $t = 0.0$    (b) $t = 0.25$    (c) $t = 0.5$    (d) $t = 0.75$    (e) $t = 1.0$    (f) Target

(g) $t = 0.0$    (h) $t = 0.25$    (i) $t = 0.5$    (j) $t = 0.75$    (k) $t = 1.0$    (l) Target

(m) $t = 0.0$    (n) $t = 0.25$    (o) $t = 0.5$    (p) $t = 0.75$    (q) $t = 1.0$    (r) Target

Figure 3: Visualization of the transformation process from the Gaussian noise to the target distribution. (a-e) are by our regularization, (g-k) are by the $L^1$ regularization, (m-q) are with the kinetic energy regularization.

an regularizer decreases the negative log-probability, denoted NLP in the table. At this experiment, 1 NFE takes approximately 0.008 seconds.

Figure 3 shows various transformation processes from the initial Gaussian noise to the target distribution with two circles. All show reasonable transformations with little differences.

The selection distribution by the auxiliary network with our regularizer is summarized in Table 9. Being similar to that in PhysioNet, DOPRI is selected for many cases, i.e., 87.5%, and the Euler method is used from time to time, i.e., 12.5%.

Table 9: The distribution of the integrator selection by our auxiliary network for Continuous Normalizing Flows

| Integrator | Percentage |
|---|---|
| DOPRI | 87.5% |
| RK4 | 0% |
| Euler | 12.5% |

## 5   CONCLUSIONS

We tackled one critical problem of Neural ODEs, a delayed process of the forward-pass inference. Even though DOPRI is able to dynamically adjust its step-sizes, as described earlier, there exists a limitation in saving the forward-pass inference time. To address the problem, we suggested i) regularizing the DOPRI's estimated error, which results in a reduced NFE, and ii) dynamically selecting an appropriate integrator for each input. We successfully showed that both the model accuracy and the forward-pass inference time can be improved at the same time in all tasks. We also showed that non-trivial percentages of test cases can be solved by the Euler method after adopting our regularizer. In particular, for MNIST our model shows more than four times smaller NFE (i.e., more than four times faster forward-pass inference) in comparison with ODE-Net without any regularizer.

One difficulty in our work is controlling hyperparameters such as $M$ and $\alpha$. If they are ill-configured, the auxiliary selection network may choose only one integrator by pursuing either computational correctness (i.e., DOPRI) or computational lightweightness (i.e., the Euler method). During our preliminary experiments, we tuned them using training data.

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
