# OpenReview forum: "DISE: Dynamic Integrator Selection to Minimize Forward Pass Time in Neural ODEs"
_ICLR.cc/2021/Conference — Reject_

### Official Review · AnonReviewer1 · 2020-10-28
**A regularization and dynamic integrator selection (DISE) are proposed to reduce NFEs at forward-pass inference. Experimental setups need to have  better explanations to clarify the effectiveness of the proposed approach.**

**Rating:** 5
**Confidence:** 4

**Review:**

Summarizing the paper claims
------------------------------------------
The paper addresses the question of reducing (on average) the number of function evaluations (NFEs) for the forward-pass inference through the neural ODE.
The proposed approach includes two main components. The first one is a direct regularization of solver's (DOPRI) estimated errors during training.  The second one is an auxiliary neural network that is learned to predict, which solver from the pre-defined set of solvers (DOPRI and fixed-step RK4, Euler) should be used during inference for a  given input sample. The paper claims that these components and their combination yield to reduce NFEs.

Strong points
-------------------
- The paper attracts attention to the fact that neural ODE architecture shouldn't stick to the sole usage of the most powerful solver during inference, depending on the input data, less powerful solvers can be applied.
- The proposed approach is evaluated on a variety of tasks (image classification, mortality prediction, continuous normalizing flows)

Weak points
-----------------
Some important details concerning the experimental setup are omitted, which makes it hard to correctly evaluate the benefits of the proposed approach and reproduce the results. Please, see below for a wider explanation.

Particularly, the following points need to be clarified to understand the fairness of the provided comparisons.

1. Were the models from the same table (Table1/Table5/Table8) trained using the same random seeds?  Neural ODE performance can significantly depend on architecture initialization, and hence using the same random seeds is required for a fair comparison.

2. Were the models from the same table (Table1/Table5/Table8) computed only once? Or provided data correspond to the mean value across several experimental runs? If the mean is provided, what is the corresponding standard deviation? Without knowing the standard deviation, it's not clear if there is a significant improvement of one method over another.

3. How many steps of RK4 and Euler are done during forward? What are hyperparameters for DOPRI (e.g., tolerance)? In the paper, I didn't find an explanation of how the number of steps for fixed-step size solvers has been picked, and how the tolerance for DOPRI has been set. The quality, as well as NFEs and DISE predictions, can vary significantly depending on these parameters.

Recommendation (accept or reject)
------------------------------------------------
For the current stage of the review, I tend to reject the paper. However, I find the topic of the paper important to the neural ODE community and will make the final score decision after the authors' clarification on crucial experimental setups.

Questions
--------------
- That would be helpful for understanding to see the test performance of pre-trained with DOPRI neural ODE when only  RK4 or only Euler is used.
- Will we observe the same behavior if we perform a comparison with adaptive methods of smaller order?
- Does the DISE strategy to choose an appropriate solver outperforms the strategy when we randomly sample solver for the next input during inference? If sampling uses the same probabilities as obtained with DISE? If uniform sampling is done?
- What is a time overhead for the training using introduced regularization? That would be nice to see plots NFEs-forward vs. epochs (wall clock time) and NFEs-backward vs. epochs (wall clock time) dependence for different methods during training

---

> ### Author Response · Authors · 2020-11-18
> **Thanks for your comments.**
>
> 1.	We will upload a revised manuscript and a revised supplementary material soon.
> 2.	NFE is much more important than the number of steps in terms of the forward pass inference time. In a step, as described in Section 2.2, DOPRI has an NFE of 6, RK4 has an NFE of 4, and the Euler method has an NFE of 1. The Euler method is at least six times faster than DOPRI even when their step sizes are identical, which is our motivation to dynamically choose between them. The total NFE of DOPRI is 6 * the number of steps whereas the total NFE of the Euler method is 1 * the number of steps. For this reason, however, there exists a limitation in decreasing NFE for DOPRI no matter how it adjusts its step size. We will make it clear in the revised manuscript. Thanks for raising this issue.
> 3.	The number of steps of RK4 and the explicit Euler method is 1 in MNIST or is decided by time in data for other datasets whereas DOPRI adaptively chooses it. The number of steps of RK4 and the explicit Euler method is a lower bound of the number of steps of DOPRI. The tolerance of DOPRI is a default value in the original Neural ODE and its related codes: 1e-3 for MNIST, 1e-5 for CNF, 1e-7 for PhysioNet. As mentioned in 1, however, the total NFE of DOPRI is 6 * the number of steps whereas the total NFE of the Euler method is 1 * the number of steps. For this reason, however, there exists a limitation in decreasing NFE for DOPRI no matter how it adjusts its step size.
> 4.	We will upload a revised manuscript with the mean and std dev values in all tables by using the same seed values.
> 5.	We tested low-order adaptive solvers. We will upload a revised supplementary material with various solvers.
> 6.	We conducted new experiments after sampling an integrator randomly or following the distributions reported in Tables 3, 6, 9.  We will upload a revised manuscript and a revised supplementary material soon.
> 7.	We prepared a chart of the wall-clock forward and backward time vs. epochs. We will upload a revised manuscript and a revised supplementary material soon.

---

> ### Author Response · Authors · 2020-11-25
> **Uploaded new version.**
>
> Thanks for your comments. We uploaded new version. Changes are highlighted in red. To minimize changes, we didn't tough minor points. We will revise them if accepted.

---

### Official Review · AnonReviewer2 · 2020-10-28
**Large changes necessary to become a clear contribution**

**Rating:** 4
**Confidence:** 3

**Review:**

The authors make two suggestions in the context of neural ODEs:
1. a regularization term based on the error estimate of an adaptive step size ODE solver (Dormand-Prince)
2. an auxiliary predictor to recommend an integrator to use for a specific sample based on minimizing the required number of function evaluations in the numerical integrator.
Based on their suggestions the authors show that it is possible to obtain improved neural ODE accuracy results at less computational cost for three tasks:
1. MNIST image classification
2. PhysioNet mortality classification
3. Continuous normalizing flows

The paper can be significantly improved in two major areas:

1. There is already important related work that the authors should take into account:

The paper "Learning differential equations that are easy to solve" (https://arxiv.org/abs/2007.04504) suggests the regularization of the k-th order derivatives with respect to time.
Based on the view of the Taylor method integrator the higher-order derivatives with respect to time are an error estimate of the current time step and also reflect on the cost of computing the solution up to a certain accuracy.

The idea in the above paper very similar to the idea of the regularization of the error estimate of an adaptive step size solver such as Dormand-Prince.
The authors say that in the Dormand-Prince method "the error is estimated by the difference between the fourth-order and the fifth-order Runge-Kutta methods".
Runge-Kutta methods use multiple (of the previous) function evaluations in order to extrapolate the solution of the next step, the higher the order, the higher the term of the Taylor expansion that is estimated (assuming the integrated function is differentiable up to the necessary order).
So the error estimation of the Dormand-Prince method is related (proportional) to a higher derivative with respect to time and regularizing it is thus very similar to the more general idea in the above paper.

The authors could make their analysis more clear and relate it to the previous work.
In general the work would benefit from a clearer exposition about adaptive step size solvers and the smoothness of the ODE at hand.

2. Principled reasoning and explanation of the auxiliary integrator recommendation system:

The purpose of an adaptive step size solver is already to make large steps where the integrated function allows this.
Given the effort it takes to properly tune the auxiliary network architecture in a task specific way it is not clear to me that there is a truly general purpose advantage (to quote the authors: "the neural network architecture for v should be carefully designed for each application").

Furthermore, the objective function of the auxiliary network is based on a discrete quantity (number of function evaluations) that is not differentiable with respect to the input.
As far as I can see the paper does not directly explain how this objective can efficiently be trained (as gradients should not be available).

I do not recommend to accept the paper since the described large changes are required for the paper to become a serious contribution.

Further recommendations:
- Give references to the claims made in the abstract already in the abstract even if the references follow in the text later. Especially for big statements like "significantly reduce number of parameters". That statement could also be improved by making it more quantitatively specific (how much is the reduction).
- Define the term "procrastinated" in the context of neural ODEs.
- The finding that "a non-trivial percentage of cases can be solved with simple integrators" seems to somewhat contradict the previous claim that "advanced integrators" have to be used. Also for "simple" (in numerical analysis terms less "stiff") cases an adaptive solver should already use much fewer steps and hence number of function evaluations.
- Introduction: two advantages... "Neural ODEs can interpret the neural network layer as a continuous variable and a hidden vector at an arbitrary layer can be calculated", why is this an advantage / what is this useful for?
- Section 2.1: The statement "It had been reported that approximating neural networks with differential equations can be done by many researchs" can be read in two ways. Maybe find a different formulation.
- Table 1: Why not also list wall-clock time of inference as that is what we are truly interested in?
- Section 2.2: "DOPRI is one of the most powerful integrators" What do you mean by powerful? How is that measured?
- Clearly explain the adaptive step size scheme of DOPRI, instead of just saying "inversely proportional to error": If I evaluate with step size h_1 and get error estimate e_1 do I then choose h_2 = 1 / e_1? How does that work exactly?
- Perhaps say something about the differentiability assumptions of the higher-order Runge-Kutta methods.
- Perhaps differentiate between explicit and implicit Euler method (instead of just saying Euler method), implicit integrators are not as unstable for stiff problems but can require many more function evaluations since they perform a nonlinear system solve at every time step.
- In Equation (2) you could make more specific what range $i$ is summed over.
- Section 3.2: "solving for h(0)", we solve with h(0) as initial data but we solve "for" h(t_final)
- How is the alpha in the exponent of the auxiliary loss chosen and for what reason?

---

> ### Author Response · Authors · 2020-11-18
> **Thanks for your comments.**
>
> 1.	We will upload a revised manuscript and a revised supplementary material soon.
> 2.	NFE is much more important than the number of steps in terms of the forward pass inference time. In a step, as described in Section 2.2, DOPRI has an NFE of 6, RK4 has an NFE of 4, and the Euler method has an NFE of 1. The Euler method is at least six times faster than DOPRI even when their step sizes are identical, which is our motivation to dynamically choose between them. The total NFE of DOPRI is 6 * the number of steps whereas the total NFE of the Euler method is 1 * the number of steps. For this reason, however, there exists a limitation in decreasing NFE for DOPRI no matter how it adjusts its step size. We will make it clear in the revised manuscript. Thanks for raising this issue.
> 3.	We are finalizing our new experiments to compare our method with the paper "Learning differential equations that are easy to solve". In fact, we already found this paper after the initial submission and have been working to revise. You can check the results after we upload a new manuscript.
> 4.	The auxiliary network is trained with NFEs, but this does not mean that its training loss is not differentiable. The ground-truth NFE values are integer values whereas the predictions of the auxiliary network can be real numbers. When the ground-truth NFE is 10, for instance, our predicted NFE can be 10.2 and the absolute error is 0.2 from which we can calculate the gradients. So, the gradient calculation is completely possible. Training the ODE and training the auxiliary network are totally separated procedures. We collect (h(0), NFE) pairs from the ODE and use these as the training data of the auxiliary network.
> 5.	We will revise following your recommendation. 1) We will add specific numbers for how many parameters can be reduced by adopting ODEs after selecting representative cases. 2) The term “procrastinated” means that the step size of DOPRI can be small and it requires long time to finish the calculation of integral. We will make it clear. 3) We will make it clear that for many cases we still need DOPRI. 4) The continuous time of Neural ODE enables us to construct continuous residual connections. We can extract hidden vectors at a continuous layer (not only from a discrete layer) and at a deep layer, e.g., a layer of 10,000 without physically constructing such a very deep network. 5) We already reported the unit NFE time so, the multiplication of the unit NFE time and the value of NFE is the wall-clock time. 6) In general, DOPRI is considered as one of the most accurate solvers producing smallest errors. 7) We will describe the step-size function of DOPRI. It is well described in the following textbook, https://www.springer.com/gp/book/9783540566700.  8) We will describe the differentiability of DOPRI. Because it consists of a series of additions at multiple different time points, it is differentiable. 9) Thanks for pointing this out. We will say the explicit Euler method in our paper. 10) The range i is the same as the number of DOPRI steps. We will make it clear. 11) We chose alpha after preliminary experiments. For some large or small alpha configuration, the auxiliary network chooses only one integrator. Our alpha is selected to make the auxiliary network work as intended.

---

> ### Author Response · Authors · 2020-11-25
> **Uploaded new version.**
>
> Thanks for your comments. Following your suggestions, we revised the paper and supplementary material. Our changes are in red. To minimize changes, we didn't tough minor points. We will revise them if accepted. The high-order regularization term shows unreliable behaviors in our experiments, e.g., runtime error and too small gradients. So, we didn't include it.

---

### Official Review · AnonReviewer4 · 2020-10-28
**Study technically sound but clarity could be improved**

**Rating:** 6
**Confidence:** 3

**Review:**

### Summary

This study proposes a method to accelerate the forward-pass in Neural ODEs, known to be a significant time bottleneck. The study is technically sound, the empirical results convincing, but the clarity could be substantially improved.


### Quality

The paper is technically sound and the claims are for the most part appropriately backed by empirical evaluation. There is just one minor point I would suggest the authors to address: the authors write "One interesting point is that RK4 had not been used at all because it does not show any notable difference from the Euler method in this task. In that regard, choosing the Euler method is a sensible decision." This claim is not really illustrated anywhere in the manuscript and it would be good if the authors show this, even if in a supplement.


### Clarity

The manuscript provides enough information for an expert reader to understand all the steps to reproduce the results. However, the text contains a substantial amount of grammar errors and imprecisions, which I would recommend the authors to tackle. Here is a (non-exhaustive) list:

-instead of "Much work has been actively studied to", "Much work has been actively devoted to";

-instead of "Neuarl ODEs and numerical methods", "Neural ODEs [...]";

-confusing formulation "It had been reported that approximating neural networks with differential equations can be done by many researchers";

-instead of "as shown in Fig. 2, consist of three parts", "as shown in Fig. 1 [...]";

-instead of "and the step size is decided by a function", "and the step size is determined by a function" or "and the step size is a function";

-instead of "Dupont et al. said that by", "Dupont et al. showed that by";

-instead of "which is not our main interest", "which is not our setting";

-instead of "Neural ODEs have one critical drawback that it requires", "Neural ODEs [...] they require";

-instead of "step size is decided by an inverse function", "step size is an inverse function";

-instead of "because the average step size will decrease", shouldn't it be "because the average step size will increase"?

-instead of "the auxiliary integrator selection network v is stabilized and we can deploy them", "the auxiliary integrator selection network [...] we can deploy it";

-confusing sentence "which is our main experimental stage". Maybe delete it for clarity?

-instead of "in average", "on average";

-instead of "in the paper or in their github repository", "in the paper or in the respective github repository";

-instead of "It is note that", "It is worth noting that";

-instead of "the task-specific loss is to maximize [...] i.e. $L_{task}$", "the task-specific loss is to maximize [...] i.e. minimize $L_{task}$".


### Originality

The novelty of the study is two fold:

(1) it proposes a regulariser to speed-up the DOPRI ODE numerical solver;
(2) it trains an auxiliary neural network to choose the most appropriate numerical solver for the Neural ODE between DOPRI, fourth-order Runge-Kutta RK4 and forward Euler.


### Significance of the work

The results suggest that the developed approach is a solid step towards developing faster Neural ODEs.

---

> ### Author Response · Authors · 2020-11-18
> **Thanks for your comments.**
>
> 1.	We will upload a revised manuscript and a revised supplementary material soon.
> 2.	In Table 6, we showed that our selection network never chooses RK4. This is because DOPRI is the most accurate method, the Euler method is the fastest method, and RK4 is in between them but we need only either DOPRI for difficult inputs or the Euler method for easy inputs. We will refer to Table 6 for our statement "One interesting point is that RK4 had not been used at all because it does not show any notable difference from the Euler method in this task. In that regard, choosing the Euler method is a sensible decision."
> 3.	Thanks for teaching us. We will accordingly revise our manuscript following your suggestions.
> 4.	To make it sure, NFE is much more important than the number of steps in terms of the forward pass inference time. In a step, as described in Section 2.2, DOPRI has an NFE of 6, RK4 has an NFE of 4, and the Euler method has an NFE of 1. The Euler method is at least six times faster than DOPRI even when their step sizes are identical, which is our motivation to dynamically choose between them. The total NFE of DOPRI is 6 * the number of steps whereas the total NFE of the Euler method is 1 * the number of steps. For this reason, however, there exists a limitation in decreasing NFE for DOPRI no matter how it adjusts its step size. We will make it clear in the revised manuscript.

---

> ### Author Response · Authors · 2020-11-25
> **Upload new version.**
>
> Thanks for your comments. We uploaded new version. Changes are highlighted in red. To minimize changes, we didn't tough minor points. We will revise them if accepted.

---

### Official Review · AnonReviewer3 · 2020-10-30
**Review of DISE**

**Rating:** 6
**Confidence:** 3

**Review:**

Summary: This paper addresses the complexity of the forward pass inference in neural ODEs. The paper proposes to augment training of the neural ODE  with an auxiliary neural network that dynamically selects the best numerical integrator for a given input sample. Furthermore, the paper also proposes a regularizer that uses the errors of the numerical integrator to reduce the number of function evaluations, without sacrificing accuracy.

The paper is well written and addresses an impediment to utilizing neural ODEs in practice. I did find the paper lacking in detail, however. For example, it is not clear where the regularizer in Eq. (2) is derived from. The authors mention a connection to the Finlay reference in Sec. 2.3, but it is not clear what this is precisely.

For the cost of each integrator in Eq. (4), how should M be chosen? What does it mean to say that a prediction is “correct”? What is the criteria being used for this purpose? It appears that the authors treat the training of the auxiliary network as a supervised learning procedure. Why is this appropriate for this task? Another way of looking at the problem is through a reinforcement learning lens, where the objective is to learn a policy mapping inputs to choices of integrators, and minimizing long-term costs (either discounted or long-term average). Of course, there is perhaps no Markov structure to the data in this setting, but presumably the inputs in the set T could be viewed as i.i.d. samples? Could the authors comment on such alternate formulations?

---

> ### Author Response · Authors · 2020-11-18
> **Thanks for your comments.**
>
> 1.	We will upload a revised manuscript and a revised supplementary material soon.
> 2.	The symbol err_{x,i} in Eq. (2) means an error at i-th DOPRI step for input x, which is the absolute difference between the 5-th order Runge-Kutta (RK5) solver and the 4-th order Runge-Kutta (RK4) solver at the step. This description is at Section 2.2 but we fail to make a connection between the symbol and the description. We will make it clear that what we described in Section 2.2 is err_{x,i}. We propose this regularization.
> 3.	We conducted a preliminary study in each dataset to decide the best value M. Since each dataset has a different range of NFEs and so on, it is needed to carefully decide M after initial investigations.
> 4.	The meaning of correct prediction is that an ODE output (with a specific integrator selected by our auxiliary selection neural network) is the same as its ground-truth value. Because all are supervised tasks in our experiments, we utilize the ground-truth information of training data.
> 5.	Thanks for pointing this out. In fact, we consider using multi-armed bandits or reinforcement learning to fine-tune the auxiliary selection network with testing data. With training data, we can collect exact training information because we know ground-truth values. So, we do not need reinforcement learning. With testing data, however, we cannot collect such information, so we plan to extend our work by adopting the direction you mentioned in the future. We expect that the selection network can be further fine-tuned with testing data if adopting multi-armed bandits or reinforcement learning. Our current plan is to use the confidence of predictions when calculating the reward from testing samples where we do not know ground-truth values.
> 6.	To make it sure, NFE is much more important than the number of steps in terms of the forward pass inference time. In a step, as described in Section 2.2, DOPRI has an NFE of 6, RK4 has an NFE of 4, and the Euler method has an NFE of 1. The Euler method is at least six times faster than DOPRI even when their step sizes are identical, which is our motivation to dynamically choose between them. The total NFE of DOPRI is 6 * the number of steps whereas the total NFE of the Euler method is 1 * the number of steps. For this reason, however, there exists a limitation in decreasing NFE for DOPRI no matter how it adjusts its step size. We will make it clear in the revised manuscript.

---

> ### Author Response · Authors · 2020-11-25
> **Uploaded new version.**
>
> Thanks for your comments. We uploaded new version. Changes are highlighted in red. To minimize changes, we didn't tough minor points. We will revise them if accepted.

---

### Decision · Program_Chairs · 2021-01-07
**Final Decision**

**Decision:**

Reject

**Comment:**

This paper proposes two methods to speed up the evaluation of neural ODEs: regularizing the ODE to be easier to integrate, and adaptively choosing which integrator to use.

These two ideas are fundamentally sensible, but the execution of the current paper is lacking.  In addition to writing and clarity issues, the main problem is not comparing to Finlay et al.  The Kelly et al paper could potentially be considered concurrent work.

I also suggest broadening the scope of the DISE method to ODE / SDE /PDE solvers in general, in situations where many similar differential equations need to be solved, amortizing the solver selection will be worthwhile even if there are no neural nets in the differential equation.

I also encourage the authors to do experiments that explore the tradeoffs of different approaches, rather than aiming just for bold lines in tables.